# Learning Plannable Representations with Causal InfoGAN

**Thanard Kurutach**[*1]     **Aviv Tamar**[*1]     **Ge Yang**[2]     **Stuart Russell**[1]     **Pieter Abbeel**[1]

## Abstract

In recent years, deep generative models have been shown to 'imagine' convincing high-dimensional observations such as images, audio, and even video, learning directly from raw data. In this work, we ask how to imagine *goal-directed visual plans* – a plausible sequence of observations that transition a dynamical system from its current configuration to a desired goal state, which can later be used as a reference trajectory for control. We focus on systems with high-dimensional observations, such as images, and propose an approach that naturally combines representation learning and planning. Our framework learns a generative model of *sequential observations*, where the generative process is induced by a transition in a low-dimensional *planning model*, and an additional noise. By maximizing the mutual information between the generated observations and the transition in the planning model, we obtain a low-dimensional representation that best explains the causal nature of the data. We structure the planning model to be compatible with efficient planning algorithms, and we propose several such models based on either discrete or continuous states. Finally, to generate a visual plan, we project the current and goal observations onto their respective states in the planning model, plan a trajectory, and then use the generative model to transform the trajectory to a sequence of observations. We demonstrate our method on imagining plausible visual plans of rope manipulation[3].

## 1   Introduction

For future robots to perform general tasks in unstructured environments such as homes or hospitals, they must be able to reason about their domain and plan their actions accordingly. In AI literature, this general problem has been investigated under two main paradigms – automated planning and scheduling [34] (henceforth, AI planning) and reinforcement learning [41] (RL).

Classical work in AI planning has drawn on the remarkable capability of humans to perform long-term reasoning and planning by using abstract representations of the world. For example, humans might think of "cup on table" as a state rather than detailed coordinates or a precise image of such a scene. Interestingly, powerful classical planners exist that can reason very effectively with these kinds of representations, as demonstrated by results in the International Planning Competition [44]. However, such logical representations of the world can be difficult to specify correctly. As an example, consider designing a logical representation for the state of a deformable object such as a rope. Moreover, logical representations that are not grounded *a priori* in real-world observation require a perception module that can identify, for example, exactly when the cup is considered "on the table".

In RL, on the other hand, a task is solved directly through trial and error, guided by a manually provided reward signal. Recent advances in model-free RL (e.g., [28, 23]) have shown remarkable success in learning policies that act directly on high-dimensional observations, such as raw images. Designing a reward function that depends on such observations can be challenging, however, and most recent studies either relied on domains where the reward can be instrumented [28, 23, 33], or

required successful demonstrations as guidance [13, 39]. Moreover, since RL is guided by the reward to solve a particular task, it does not automatically generalize to different tasks [43, 18].

In principle, model-based RL can solve the generalization problem by learning a dynamics model and planning with that model. However, applying model-based RL to domains with high-dimensional observations has been challenging [46, 14, 12]. Deep learning approaches to learning dynamics models (e.g., action-conditional video prediction models [30, 1, 12]) tend to get bogged down in pixel-level detail, tend to be computationally expensive, and are far from accurate over longer time scales. Moreover, the representations learned using such approaches are typically unstructured, high-dimensional continuous vectors, which cannot be used in efficient planning algorithms.

In this work, we aim to combine the merits of deep learning dynamics models and classical AI planning, and propose a framework for long-term reasoning and planning that is grounded in real-world perception. We present **Causal InfoGAN (CIGAN)**, a method for learning *plannable representations* of dynamical systems with high-dimensional observations such as images. By plannable, we mean representations that are structured in such a way that makes them amenable for efficient search, through AI planning tools. In particular, we focus on discrete and deterministic dynamics models, which can be used with graph search methods, and on continuous models where planning is done by linear interpolation, though our framework can be generalized to other model types.

In our framework, a generative adversarial net (GAN; [15]) is trained to generate sequential observation pairs from the dynamical system. The GAN generator takes as input both unstructured random noise and a structured pair of consecutive states from a low-dimensional, parametrized dynamical system termed the *planning model*. The planning model is meant to capture the features that are *most essential for representing the causal properties* in the data, and are therefore important for planning future outcomes. To learn such a model, we follow the InfoGAN idea [5], and add to the GAN training loss a term that maximizes the mutual information between the observation pairs and the transitions that induced them.

The CIGAN model can be trained using random exploration data from the system. After learning, given an observation of an initial configuration and a goal configuration, it can generate a "walkthrough" sequence of feasible observations that lead from the initial state to the goal. This walkthrough can be later used as a reference signal for a controller to execute the task in the real system. We demonstrate convincing walkthrough generation on synthetic tasks and real image data collected by Nair et al. [29] of a robot randomly poking a rope.

## 2   Preliminaries and Problem Formulation

Let $H$ denote the entropy of a random variable, and $I$ denote the mutual information between two random variables [8].

**GAN and InfoGAN:** Deep generative models aim to generate samples from the real distribution of observations, $P_{\text{data}}$. In this work we build on the GAN framework [15], which is composed of a generator, $G(z) = o$, mapping a noise input $z \sim P_{\text{noise}}(z)$ to an observation $o$, and a discriminator, $D(o)$, mapping an observation to the probability that it was sampled from the real data. The GAN training optimizes a game between the generator and discriminator,

$$\min_{G} \max_{D} V(G, D) = \min_{G} \max_{D} \mathbb{E}_{o \sim P_{\text{data}}} \left[ \log D(o) \right] + \mathbb{E}_{z \sim P_{\text{noise}}} \left[ \log \left( 1 - D(G(z)) \right) \right].$$

One can view the noise vector $z$ in the GAN as a representation of the observation $o$. In GAN training, however, there is no incentive for this representation to display any structure at all, making it difficult to interpret, or use in a downstream task. The InfoGAN method [5] aims to mitigate this issue. The idea in InfoGAN is to add to the generator input an additional 'state'[4] component $s \sim P(s)$, and add to the GAN objective a loss that induces maximal mutual information between the generated observation and the state. The InfoGAN objective is given by:

$$\min_{G} \max_{D} \quad V(G, D) - \lambda I\left(s; G(z, s)\right), \tag{1}$$

where $\lambda > 0$ is a weight parameter, and $V(G, D)$ is the GAN loss above. Intuitively, this objective induces the state to capture the most salient properties of the observation. Optimizing the objective in (1) directly is difficult without access to the posterior distribution $P(s|o)$, and a variational lower bound was proposed in [5]. Define an auxiliary distribution $Q(s|o)$ to approximate the posterior $P(s|o)$. Then, $I\left(s; G(z, s)\right) \geq \mathbb{E}_{s \sim P(s), o \sim G(z,s)} \left[ \log Q(s|o) \right] + H(s)$. Using this bound, the InfoGAN objective (1) can be optimized using stochastic gradient descent.

**Problem Formulation:** We consider a fully observable and deterministic dynamical system, $o_{t+1} = f(o_t, u_t)$, where $o_t$ and $u_t$ denote the observation and action at time $t$, respectively. The function $f$ is assumed to be unknown. We are provided with data $\mathcal{D}$ in the form of $N$ trajectories of observations $\left\{ o_1^i, u_1^i \dots, o_{T_i}^i \right\}_{i \in 1, \dots, N}$, generated from $f$, where the actions are generated by an arbitrary exploration policy.[5]

We say that two observations $o, o'$ are *h-reachable* if there exists a sequence of actions that takes the system from $o$ to $o'$ within $h$ steps or less. We consider the problem of generating a *walkthrough* – a sequence of reachable observations along a feasible path between the start and the goal:

**Problem 1** *Walkthrough Planning: Given $\mathcal{D}$, $h$, and two observations $o_{start}, o_{goal}$, generate a sequence of observations $o_{start}, \dots, o_{goal}$ such that every two consecutive observations in the sequence are h-reachable. If such a sequence does not exist, return $\emptyset$.*

The motivation to solve problem 1 is that it breaks the long horizon planning problem (from $o_{start}$ to $o_{goal}$) into a sequence of short $h$-horizon planning problems which can be later solved effectively using other methods such as inverse dynamics or model-free RL [29]. This concept of *temporal abstraction* has been fundamental in AI planning (e.g., [11, 42]). Since we are searching for a sequence of way point observations, the actions are not relevant for our problem, and in the sequel we omit them from the discussion.

## 3 Causal InfoGAN

A natural approach for solving the walkthrough planning problem in Section 2 is learning some model of the dynamics $f$ from the data, and searching for a plan within that model. This leads to a trade-off. On the one hand, we want to be expressive, and learn all the transitions possible from every $o$ within a horizon $h$. When $o$ is a high dimensional image observation, this typically requires mapping the image to an extensive feature space [30, 12]. On the other hand, however, we want to plan efficiently, which generally requires either low dimensional state spaces or well-structured representations. We approach this challenge by proposing *Causal InfoGAN* – an expressive generative model with a structured representation that is compatible with planning algorithms. In this section we present the Causal InfoGAN generative model, and in Section 4 we explain how to use the model for planning.

Let $o$ and $o'$ denote a pair of sequential observations from the dynamical system $f$, and let $P_{\text{data}}(o, o')$ denote their probability, as displayed in the data $\mathcal{D}$. We posit that a generative model that can accurately learn $P_{\text{data}}(o, o')$ has to capture the features that are important for representing the *causality* in the data – what next observations $o'$ are reachable from the current observation $o$. Naturally, such features would be useful later for planning.

We build on the GAN framework [15]. Applied to our setting, a vanilla GAN would be composed of a generator, $o, o' = G(z)$, mapping a noise input $z \sim P_{\text{noise}}(z)$ to an observation pair, and a discriminator, $D(o, o')$, mapping an observation pair to the probability that it was sampled from the real data $\mathcal{D}$ and not from the generator. One can view the noise vector $z$ in such a GAN as a feature vector, containing some representation of the transition to $o'$ from $o$. The problem, however, is that the structure of this representation is not necessarily easy to decode and use for planning. Therefore, we propose to design a generator with a *structured* input that can be later used for planning. In particular, we propose a GAN generator that is driven by states sampled from a parametrized dynamical system.

Let $\mathcal{M}$ denote a dynamical system with state space $S$, which we term the set of *abstract-states*, and a parametrized, stochastic transition function $T_{\mathcal{M}}(s'|s)$: $s' \sim T_{\mathcal{M}}(s'|s)$, where $s, s' \in S$ are a pair of consecutive abstract states. We denote by $P_{\mathcal{M}}(s)$ the prior probability of an abstract state $s$. We emphasize that the abstract state space $S$ can be different from the space of real observations $o$. For reasons that will become clear later on, we term $\mathcal{M}$ as the *latent planning system*.

We propose to structure the generator as taking in a pair of consecutive abstract states $s, s'$ in addition to the noise vector $z$. The GAN objective in this case is therefore (cf. Section 2):

$$V(G, D) = \mathbb{E}_{o,o' \sim P_{\text{data}}} \left[ \log D(o, o') \right] + \mathbb{E}_{z \sim P_{\text{noise}}, s \sim P_{\mathcal{M}}(s), s' \sim T_{\mathcal{M}}(s)} \left[ \log \left( 1 - D(G(z, s, s')) \right) \right]. \quad (2)$$

The idea is that $s$ and $s'$ would represent the abstract features that are important for understanding the causality in the data, while $z$ would model variations that are less informative, such as pixel level details. To learn such representations, we follow InfoGAN [5], and add to the GAN objective a term that maximizes mutual information between the generated pair of observations and the abstract states.

We propose the Causal InfoGAN objective:

$$\min_{\mathcal{M},G} \max_D \quad V(G,D) - \lambda I\left(s,s';o,o'\right), \quad \text{s.t. } o,o' \sim G(z,s,s'); \quad s \sim P_{\mathcal{M}}; \quad s' \sim T_{\mathcal{M}}(s), \ (3)$$

where $\lambda > 0$ is a weight parameter, and $V(G,D)$ is given in (2). Intuitively, this objective induces the abstract model to capture the most salient possible changes that can be effected on the observation.

Optimizing the objective in (3) directly is difficult, since we do not have access to the posterior distribution, $P(s,s'|o,o')$, when using an expressive generator function. Following InfoGAN [5], we optimize a variational lower bound of (3). Define an auxiliary distribution $Q(s,s'|o,o')$ to approximate the posterior $P(s,s'|o,o')$. We have, following a similar derivation to [5]:

$$I\left((s,s');G(z,s,s')\right) \geq \mathbb{E}_{\substack{s \sim P_{\mathcal{M}}, s' \sim T_{\mathcal{M}}(s), \\ o,o' \sim G(z,s,s')}} \left[\log Q(s,s'|o,o')\right] + H(s,s') \doteq I_{VLB}(G,Q). \quad (4)$$

To encourage the same mapping between $s,o$ and $s',o'$, we propose the disentangled posterior approximation, $Q(s,s'|o,o') = Q(s|o)Q(s'|o')$ (see Appendix B.)

We plug the lower bound (4) in Eq. (3) to obtain the following loss function:

$$\min_{G,Q,\mathcal{M}} \max_D \quad V(G,D) - \lambda I_{\text{VLB}}(G,Q), \quad\quad\quad (5)$$

where $\lambda > 0$ is a constant. The loss in (5) can be optimized effectively using stochastic gradient descent, and we provide a detailed algorithm in Appendix C.

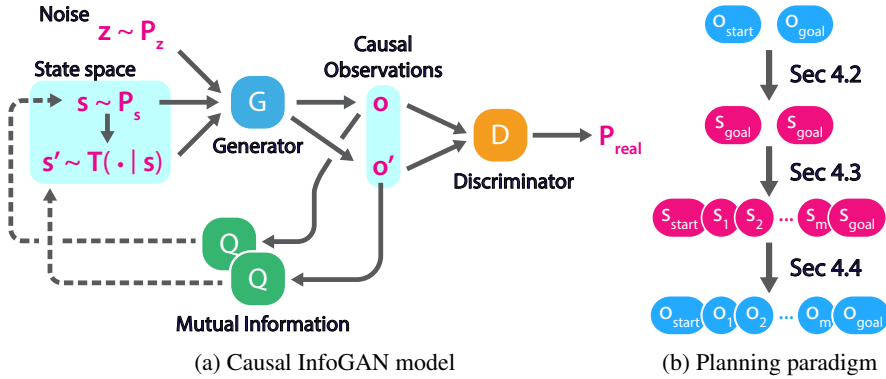

(a) Causal InfoGAN model          (b) Planning paradigm

Figure 1: The Causal InfoGAN framework. **(a) Generative model (cf. Section 3)**. First, an abstract state $s$ is sampled from a prior $P_{\mathcal{M}}(s)$. Given $s$, the next state $s'$ is sampled using the transition model $T_{\mathcal{M}}(s'|s)$. The states $s, s'$ are fed, together with a random noise sample $z$, into the generator which outputs $o, o'$. The discriminator $D$ maps an observation pair to the probability of the pair being real. Finally, the approximate posterior $Q$ maps from each observation to the distribution of the state it associates with. The causal InfoGAN loss function in Equation (5) encourages $Q$ to predict each state accurately from each observation. **(b) Planning paradigm (cf. Section 4)**. Given start and goal observations, we first map them to abstract states, and then we apply planning algorithms using the model $\mathcal{M}$ to search for a path from $s_{start}$ to $s_{goal}$. Finally, from the plan in abstract states, we generate back a sequence of observations.

## 4 Planning with Causal InfoGAN models

In this section, we discuss how to use the Causal InfoGAN model for planning goal directed trajectories. We first present our general methodology, and then propose several model configurations for which (5) can be optimized efficiently, and the latent planning system is compatible with efficient planning algorithms. We then describe how to combine these ideas for solving the walkthrough planning problem in various domains.

### 4.1 General Planning Paradigm

Our general paradigm for goal directed planning is described in Figure 1b. We start by training a Causal InfoGAN model from the data, as described in the previous section. Then, we perform the following 3 steps, which are detailed in the rest of this section:

1. Given a pair of observations $o_{start}, o_{goal}$, we first encode them into a pair of corresponding states $s_{start}, s_{goal}$. This is described in Section 4.2.

| Type | Values $s$ | Prior $P_{\mathcal{M}}(s)$ | Transition $T_{\mathcal{M}}(s'|s)$ | Planning algorithms |
|---|---|---|---|---|
| Discrete – one-hot | $[N]$ | $\mathcal{U}\{1,\ldots,N\}$ | $s' \sim Softmax(s^{\top}\theta)$ | Dijkstra |
| Discrete – binary | $\{0,1\}^N$ | $\mathcal{U}\{0,1\}^N$ | See eq. 6 | Dijkstra |
| Continuous | $\mathbb{R}^N$ | $\mathcal{U}(-1,1)^N$ | $s' \sim \mathcal{N}(s, \Sigma_\theta(s))$ | Linear interpolation |

Table 1: Various latent planning systems. In all cases, $N$ is the state dimension. The parameters $\theta$ of the transition $T_{\mathcal{M}}$ depending on the state type. In the one-hot case, $\theta$ is a matrix in $\mathbb{R}^{N \times N}$. In the binary case, $\theta$ denotes parameters in a stochastic neural network; see Eq. (6). In the continuous case $\theta$ represents the parameters of a neural network that controls the variance of the transition.

2. Using the transition probabilities in the planning model $\mathcal{M}$, we plan a feasible state trajectory from $s_{start}$ to $s_{goal}$: $s_{start}, s_1, \ldots, s_m, s_{goal}$. This is described in Section 4.3.

3. Finally, we decode the state trajectory into a corresponding trajectory of observations $o_{start}, o_1, \ldots, o_m, o_{goal}$. This is described in Section 4.4.

The specific method for each step in the planning paradigm can depend on the problem at hand. For example, some systems are naturally described by discrete abstract states, while others are better described by continuous states. In the following, we describe several models and methods that worked well for us, under the general planning paradigm described above. This list is by no means exhaustive. On the contrary, we believe that the Causal InfoGAN framework provides a basis for further investigation of deep generative models that are compatible with planning.

## 4.2 Encoding an Observation to a State

For mapping an observation to a state, we can simply use the disentangled posterior $Q(s|o)$. We found this approach to work well in low-dimensional observation spaces. However, for high-dimensional image observations we found that the learned $Q(s|o)$ was accurate in classifying generated observations (by the generator), but inaccurate for classifying real observations. This is explained by the fact that in Causal InfoGAN, $Q$ is only trained on generated observations, and can overfit to generated images. For images, we therefore opted for a different approach. Following [45], we performed a search over the latent space to find the best latent state mapping $s^*(o)$: $s^*(o) = \arg\min_s \min_{s',z} \|o - G(s, s', z)\|^2$. Another approach, which could scale to complex image observations, is to add to the GAN training an explicit encoding network [10, 49]. In our experiments, the simple search approach worked well and we did not require additional modifications to the GAN training.

## 4.3 Latent Planning Systems

We now present several latent planning systems that are compatible with efficient planning algorithms. Table 1 summarizes the different models. In all cases, optimizing the model parameters with respect to the expectation in the loss (4) is done using the reparametrization trick (following [20] for continuous states, and [16] for discrete states).

**Discrete Abstract States – One-Hot Representation.** We start from a simple abstract state representation, in which each $s \in S$ is represented as a $N$−dimensional one-hot vector. We denote by $\theta \in \mathbb{R}^{N \times N}$ the model parameters, and compute transition probabilities as: $T_{\mathcal{M}}(s'|s) = Softmax(s^{\top}\theta)$.

**Discrete Abstract States – Binary Representation.** We present a more expressive abstract state representation using binary states. Binary state representations are common in AI planning, where each binary element is known as a *predicate*, and corresponds to a particular object property being true or false [34]. Using Causal InfoGAN, we learn the predicates *directly from data*.

We propose a parametric transition model that is suitable for binary representations. Let $s \in \{0,1\}^N$ be an $N$−dimensional binary vector, drawn from $P_{\mathcal{M}}(s)$. We generate the next state $s'$ by first drawing a random *action vector* $a \in \{0,1\}^M$ with some probability $P_{\mathcal{M}}(a)$ [6]. Let $l_i$ denote a feed-forward neural network with sigmoid output parametrized by $\theta$ that maps the state $s$ and action $a$ to the Bernoulli's parameter of $s'_i|s, a$. Thus, the probability of the next state $s'$ is finally given by:

$$T_{\mathcal{M}}(s' = v|s) = \mathbb{E}_a\left[\prod_i T_{\mathcal{M}}(s'_i = v_i|s, a)\right] = \mathbb{E}_a\left[\prod_i l_i(s,a)^{v_i}(1 - l_i(s,a))^{(1-v_i)}\right]. \quad (6)$$

We emphasize that there is not necessarily any correspondence between the action vector $a$ and the real actions that generated the observation pairs in the data. The action $a$ is simply a means to induce stochasticity to the state transition network.

For planning with discrete models, we interpret the stochastic transition model $T_{\mathcal{M}}$ as providing the possible state transitions, i.e., for every $s'$ such that $T_{\mathcal{M}}(s'|s) > \epsilon$ there exists a possible transition from $s$ to $s'$. For planning, we require abstract state representations that are compatible with efficient AI planning algorithms. The one-hot and binary representations above can be directly plugged in to graph-planning algorithms such as Dijkstra's shortest-path algorithm [34].

**Continuous Abstract States.** For some domains, such as the rope manipulation in our experiments, a continuous abstract state is more suitable. We consider a model where an $s \in S$ is an $N-$dimensional continuous vector. Planning in high-dimensional continuous domains, however, is hard in general.

Here, we propose a simple and effective solution: we will learn a latent planning system such that *linear interpolation between states makes for feasible plans*. To bring about such a model, we consider transition probabilities $T_{\mathcal{M}}(s'|s)$ given as Gaussian perturbations of the state: $s' = s + \delta$, where $\delta \sim \mathcal{N}(0, \Sigma_\theta(s))$ and $\Sigma_\theta$ is a diagonal covariance matrix, represented by a MLP with parameters $\theta$. The key idea here is that, if only small local transitions are possible in the system, then a linear interpolation between two states $s_{start}, s_{goal}$ has a high probability, and therefore represents a feasible trajectory in the observation space. To encourage such small transitions, we add an L2 norm of the convariance matrix to the loss function (5): $L_{cont}(\mathcal{M}) = \mathbb{E}_{s \sim P_{\mathcal{M}}} \|\Sigma_\theta(s)\|_2$. The prior probability $P_{\mathcal{M}}$ for each element of $s$ is uniform in $[-1, 1]$.

## 4.4 Decoding a State Trajectory to an Observation Walkthrough Trajectory

We now discuss how to generate a feasible sequence of observations from the planned state trajectory. Here, as before, we separate the discussion for systems with low-dimensional observations and systems with image observations, as we found that different practices work best for each.

For low-dimensional observations, we structure the GAN generator $G$ to have an observation-conditional form: $o = G_1(z, s, s')$, $o' = G_2(z, o, s, s')$. Using this generator form, we can sequentially generate observations from a state sequence $s_1, \ldots, s_T$. We first use $G_1$ to generate $o_1$ from $s_1, s_2$, and then, for each $2 \leq t < t$, use $G_2$ to generate $o_{t+1}$ from $s_t, s_{t+1}$, and $o_t$.

For high-dimensional image observations, the sequential generator does not work well, since small errors in the image generation tend to get accumulated when fed back into the generator. We therefore follow a different approach. To generate the $i$'th observation in the trajectory $o_i$, we use the generator with the input $s_i, s_{i+1}$, and a noise $z$ that is fixed throughout the whole trajectory. The generator actually outputs a pair of sequential images, but we discard the second image in the pair.

To further improve the planning result we generate $K$ random trajectories with different random noise $z$, and select the best trajectory by using a discriminator $D$ to provide a confidence score for each trajectory. In the low-dimensional case, we use the GAN discriminator. In the high-dimensional case, however, we find that the discriminator tends to overfit to the generator. Therefore, we trained an auxiliary discriminator for novelty detection, as described in the Experiment Section 6.2.

## 5 Related Work

Combining deep generative models with structured dynamical systems has been explored in the context of variational autoencoders (VAEs), where the latent space was continuous [6, 17]. Watter et al. [46] used such models for planning, by learning latent linear dynamics, and using a linear quadratic Gaussian control algorithm. Disentangled video prediction [9] separates object content and position, but has not been used for planning. Very recently, Corneil et al. [7] suggested Variational State Tabulation (VaST) – a VAE-based approach for learning latent dynamics over binary state representations, and planning in the latent space using prioritized sweeping to speed up RL. Causal InfoGAN shares several similarities with VaST, such as using Gumbel-Softmax to backprop through transitions of discrete binary states, and leveraging the structure of the binary states for planning. However, VaST is formulated to require the agent actions, and is thus limited to single time step predictions. More generally, our work is developed under the GAN formulation, which, to date, has several benefits over VAEs such as superior quality of image generation [19]. Causal InfoGAN can also be used with continuous abstract states.

The semiparametric topological memory (SPTM) [37] is another recent approach for solving problems such as Problem 1, by planning in a graph where every observation in the data is a node, and connectivity is decided using a learned similarity metric between pairs of observations. SPTM has shown impressive results on image-based navigation. However, Causal InfoGAN's *parametric approach* of learning a compact, model for planning has the potential to scale up to more complex problems, in which the increasing amount of data required would make the nonparametric SPTM approach difficult to apply.

Learning state aggregation and state representation has a long history in RL. Methods such as in [27, 38] exploit the value function for measuring state similarity, and are therefore limited to the task defined by the reward. Methods for general state aggregation have also been proposed, based on spectral clustering [26, 40, 25, 24], and variants of K-means [3]. All these approaches rely in some form on the Euclidean distance as a metric between observation features. As we show in our experiments, the Euclidean distance can be unsuitable even on low-dimensional continuous domains. Recent work in deep RL explored learning goal-conditioned value functions and policies [2, 32], and policies with an explicit planning computation [43, 31, 39]. These approaches require a reward signal for learning (or supervision from an expert [39]). In our work, we do not require a reward signal, and learn a general model of the dynamical system, which is used for goal-directed planning.

Our work is also related to learning models of intuitive physics. Previous work explored feedforward neural networks for predicting outcomes of physical experiments [22], neural networks for modelling relations between objects [47, 36], and prediction based on physics simulators [4, 48]. To the best of our knowledge, these approaches cannot be used for planning. However, related ideas would likely be required for scaling our method to more complex domains, such as manipulating several objects.

In the planning literature, most studies relied on manually designed state representations. In a recent work, Konidaris et al. [21] automatically extracted state representations from raw observations, but relied on a prespecified set of skills for the task. In our work, we automatically extract state representations by learning salient features that describe the causal structure of the data.

# 6 Experiments

In our experiments, we aim to (1) visualize the abstract states and planning in CIGAN; (2) compare CIGAN with recent state-aggregation methods in the literature; (3) show that CIGAN can produce realistic visual plans in a complex dynamical system; and (4) show that CIGAN significantly outperforms baseline methods. We begin our investigation with a set of toy tasks, specifically designed to demonstrate the benefits of CIGAN, where we can also perform an extensive quantitative evaluation. We later present experiments on a real dataset of robotic rope manipulation.

## 6.1 Illustrative Experiments

In this section we evaluate CIGAN on a set of 2D navigation problems. These problems abstract away the challenges of learning visual features, allowing an informative comparison on the task of learning causal structure in data, and using it for planning. For details of the training data see Appendix **??**.

Our toy domains involve a particle moving in a 2-dimensional continuous domain with impenetrable obstacles, as depicted in Figure 2, and Figure 5. in Appendix A.1. The observations are the $(x, y)$ coordinates of the particle in the plane, and, in the door-key domain, also a binary indicator for holding the key. We generate data trajectories by simulating a random motion of the particle, started from random initial points. We consider the following various geometrical arrangements of the domain, chosen to demonstrate the properties of our method.

1. **Tunnels:** the domain is partitioned into two unconnected rooms (top/bottom), where in each room there is an obstacle, positioned such that transitioning between the left/right quadrants is through a narrow tunnel.

2. **Door-key:** two rooms are connected by a door. The door can be traversed only if the agent holds the key, which is obtained by moving to the red-marked area in the top right corner of the upper room. Holding the key is represented as a binary 0/1 element in the observation.

3. **Rescaled door-key:** Same as door key domain, but the key observation is rescaled to be a small $\epsilon$ when the agent is holding the key, and 0 otherwise.

Our domains are designed to distinguish when standard state aggregation methods, which rely on the Euclidean metric, can work well. In the tunnel domain, the Euclidean metric is not informative about the dynamics in the task – two points in different rooms inside the tunnel can be very close in Euclidean distance, but not connected, while points in the same room can be more distant but connected. In the door-key domain, the Euclidean distance is informative if observations with key and without key are very distant in Euclidean space, as in the 0/1 representation (compared to the domain size which is in $[-1, 1]$). In the rescaled door-key, we make the Euclidean distance less informative by changing the key observation to be $0/\epsilon$.

We compare CIGAN with several recent methods for aggregating observation features into states for planning. Note that in these simple 2D domains, feature extraction is not necessary as the observations are already low dimensional vectors. The simplest baseline is K-means, which relies on the Euclidean distance between observations. In [3], a variant of K-means for temporal data was proposed, using a

window of consecutive observations to measure a smoothed Euclidean distance to a cluster centroids. We refer to this method as temporal K-means. In [26], and more recently [40] and [25], spectral clustering (SC) was used to cluster observations. For continuous observations, SC requires a distance function to build a connectivity graph, and previous studies [26, 40, 25] relied on the Euclidean distance, using either nearest neighbor to connect nodes, or exponentiated-distance weighted edges.

In Figure 2, we show the CIGAN classification of observations to abstract states, $Q(s|o)$, and compare with the K-means baseline; the other baselines gave qualitatively similar results. Note that CIGAN learned a clustering that is related to the dynamical properties of the domain, while the baselines, which rely on a Euclidean distance, learned clusters that are not informative about the real possible transitions. As a result, CIGAN clearly separates abstract states within each room, while the K-means baseline clusters observations across the wall. This demonstrates the potential of CIGAN to learn meaningful state abstractions without requiring a distance function in observation space. Similarly, the results for door-key domains are shown in Appendix A.1.

To evaluate planning performance, we hand-coded an oracle function that evaluates whether an observation trajectory is feasible or not (e.g., does not cross obstacles, correctly reports $\emptyset$ when a trajectory does not exist). For CIGAN, we ran the planning algorithm described in Section 4. For baselines, we calculated cluster transitions from the data, and generated planning trajectories in observation space by using the cluster centroids. We chose algorithm parameters and stopping criteria by measuring the average feasibility score on a validation set of start/goal observations, and report the average feasibility on a held out test set of start/goal observations. Our results in Table 2 show that by learning more informative clusters, CIGAN resulted in significantly better planning.

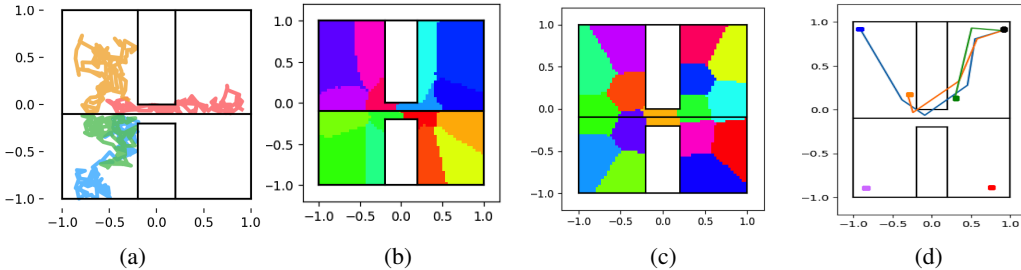

(a)          (b)          (c)          (d)

Figure 2: 2D particle results on tunnel domain. (a) The domain - top/bottom rooms are not connected. Left/right quadrants are connected through a narrow tunnel. An example of several random walk trajectories are shown. (b) Clustering found by CIGAN. (c) Clustering found by K-means. (d) Example walkthrough trajectories generated by CIGAN, from a point at the top right to five other locations on the map, marked by colored circles. For trajectories that were not found only the target is shown. Note that CIGAN learned clusters that correspond to the possible dynamics of the particle in the task, and was therefore able to generate reasonable planning trajectories.

|  | Tunnels | Door-key | Rescaled door-key |
|---|---|---|---|
| CIGAN | **98%** | 98% | **97%** |
| K-means | 12.25% | **100%** | 0.0% |
| Temporal K-means | 7.0% | **100%** | 0.0% |
| Spectral clustering | 8.75% | 60% | 20.0% |

Table 2: Planning results for 2D tasks. Table shows average feasibility of plans (higher is better) generated by the different algorithms. Note that CIGAN significantly outperforms baselines in domains where the Euclidean distance is not informative for planning.

## 6.2 Rope Manipulation

In this section we demonstrate CIGAN on the task of generating realistic walkthroughs of robotic rope manipulation. Then, we show that CIGAN generates significantly better trajectories than those generated by the state-of-the-art generative model baselines both visually and quantitatively.

The rope manipulation dataset [29] contains sequential images of rope manipulated in a self-supervised manner, by a robot randomly choosing a point on the rope and perturbing it slightly. Using these data, the task is to manipulate the rope in a goal-oriented fashion, from one configuration to another, where a goal is represented as an image of the desired rope configuration. In the original study, Nair et al. [29] used the data to learn an inverse dynamics model for manipulating the rope between two images of similar rope configurations. Then, to solve long-horizon planning, Nair

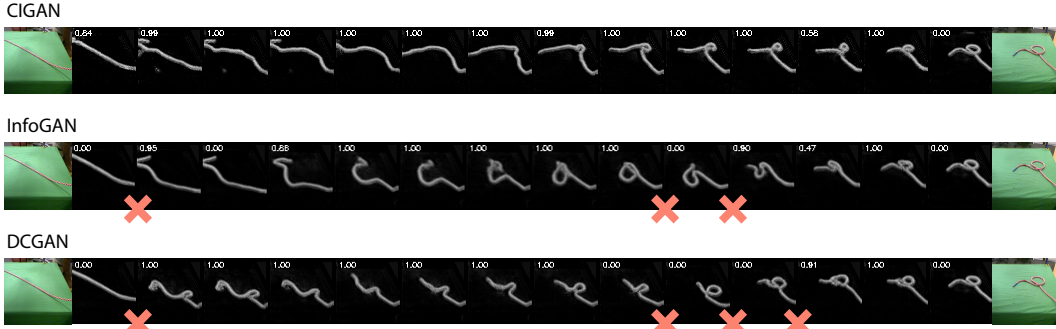

Figure 3: Rope walkthroughs generated from CIGAN, InfoGAN, and DCGAN. Red crosses show unfeasible one-step transitions with respect to the data. See more plans in Appendix A.2.

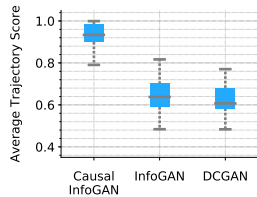

Figure 4: Evaluation of walkthrough planning in rope domain. We trained a classifier to predict whether two observations are sequential or not (1=sequential, 0=not sequential), and compare the average classification score for different generative models. Note that CIGAN significantly outperforms the baselines, in alignment with the qualitative results of Figure 3.

et al. required a human to provide the walkthrough sequence of rope poses, and used the learned controller to execute the short-horizon transitions within the plan. In our experiment, we show that CIGAN can be used to generate walkthrough plans *directly from data* for long-horizon tasks, without requiring additional human guidance. We train a CIGAN model on the rope manipulation data of [29]. We pre-processed the data by removing the background, and applying a grayscale transformation. We chose the continuous abstract state representation with the linear interpolation planner as described in Section 4. Note that, as described in Section 4, the encoding, planning, and decoding methods in this case are not specific to CIGAN, and can be used with a GAN or InfoGAN generative model, allowing a fair comparison with alternative representation learning methods. In Figure 3, we generate walkthroughs using CIGAN, InfoGAN, and DCGAN Noticeably, Causal InfoGAN resulted in a smooth latent space where linear interpolation indeed corresponds to plausible trajectories.

To numerically evaluate planning performance we propose a visual fidelity score, inspired by the Inception score for evaluating GANs [35], we train a binary classifier to classify whether two images are sequential in the data or not[7]. For an image pair, the classifier output therefore provides a score between 0 and 1 for the feasibility of the transition. We then compute the *trajectory score* – the average classifier score of image pairs in the trajectory. Note that this classifier is trained independent of the generative models, making for an impartial metric. For each start and goal, we pick the best trajectory score out of 400 samples of the noise variable $z$.[8] As shown in Figure 4, Causal InfoGAN achieved a significantly higher trajectory score averaged over 57 task configurations.

## 7 Conclusion

We presented Causal InfoGAN, a framework for learning deep generative models of sequential data with a structured latent space. By choosing the latent space to be compatible with efficient planning algorithms, we developed a framework capable of generating goal-directed trajectories from high-dimensional dynamical systems.

Our results for generating realistic manipulation plans of rope suggest promising applications in robotics, where designing models and controllers for manipulating deformable objects is challenging.

The binary latent models we explored provide a connection between deep representation learning and classical AI planning, where Causal InfoGAN can be seen as a method for learning object predicates directly from data. In future work we intend to investigate this direction further, and incorporate object-oriented models, which are a fundamental component in classical AI.

## Acknowledgement

This work was funded in part by ONR PECASE N000141612723 and Siemens. Thanard Kurutach and Aviv Tamar were supported by ONR PECASE N000141612723. Aviv Tamar was partially supported by the Technion Viterbi scholarship. The authors wish to thank Tom Zahavy and Aravind Srinivas for sharing their code for our comparisons with MDP state aggregation baselines, and Elinor Tamar for timely assistance in preprocessing the rope data.

## Footnotes

[1]Berkeley AI Research, University of California, Berkeley

[2]Department of Physics, University of Chicago

[3]Code is available online at `http://github.com/thanard/causal-infogan`.

[4]In [5], $s$ is referred to as a *code*. Here we term it as a state, to correspond with our subsequent development of structured GAN input from a dynamical system.

[5]In this work, we do not concern the problem of how to best generate the exploration data.

[6]See Appendix B.2 Binary States for experimental choices.

[7]The positive data are the pairs of rope images that are 1 step apart and the negative data are randomly chosen pairs that are from different runs which are highly likely to be farther than 1 step apart.

[8]This selection process is applied the same way to the DCGAN and InfoGAN baselines.

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
