[Supplementary Material · NIPS18___Camera_ready___Causal_InfoGAN__ supplementary.pdf]

# A Additional results

## A.1 Door-key Results

In Figure 5 we show similar results for the door-key domain. When the key observation was scaled to $0/\epsilon$, standard clustering methods did not separate states with the key and without key to different clusters. CIGAN, on the other hand, learned a binary predicate for holding the key, and learned that obtaining the key happens in the correct position.

|  (a) | (b) | (c) | (d) |

Figure 5: 2D particle results on the $\epsilon$-key domain where the key dimension is scaled down to 0.1. (a) **The key domain**: The rooms are separated in-between by a wall with a door (yellow). The door only opens when the agent has the key, which can be obtained if the agent is within the area indicated by the red circle on the upper right corner. (b) From no-key to has key, **k-means**: Value indicates the probability for the agent to transition from a state not having the key to a state having the key at each $(x, y)$ location. This transition should only occur near the key region (indicated by the red ring). In this case, K-means fails to learn the separation between having and not having the key, and generated high transition probability over the entire domain. (c) The same figure as (b), generated by **Causal InfoGAN**. On the top right corner where the key is located, the GAN correctly learns that it can transition from having no key to having the key. Bottom blots appear where the posterior sees no data. (d) Causal-InfoGAN planned walkthrough trajectories, showing how the agent acquires the key to cross the door. When the goal is in the top room, the agent goes directly towards the goal without making a detour for the key region.

## A.2 Rope Results

In Figure 6, we provide additional planning results generated by Causal InfoGAN.

# B Detailed description of our approach

## B.1 Disentangled Posterior Approximation

We now note a subtle point. The mutual information in (3) is not sensitive to the order of the code words of the random variables $s$ and $s'$.[9] This points to a potential caveat in the optimization objective (3): we would like the random variable for the next abstract state $s'$ to have the *same meaning* as the random variable for the abstract state $s$. That would allow us to roll-out a sequence of changes to the abstract state, by applying the transition operator $T_{\mathcal{M}}$ sequentially, and effectively plan in the abstract model $\mathcal{M}$. However, the loss function (3) does not automatically induce $s'$ to have the same meaning as $s$, since We solve this problem by proposing the *disentangled posterior approximation*, $Q(s, s'|o, o') = Q_1(s|o)Q_2(s'|o')$, and choose $Q_1 = Q_2 \doteq Q$. This effectively induces a generator for which $P(s|o) = P(s'|o')$.[10]

(a) Causal InfoGAN

(b) InfoGAN

(c) DCGAN

Figure 6: Results for rope manipulation data. We compare planning using Causal InfoGAN (top), InfoGAN (middle), and DCGAN (bottom) by interpolation in the latent space, for several rope manipulation goals starting from the same initial configuration. Each plot shows 5 planning instances, from left (starting observation) to the right (goal observation). For each instance, the shown trajectory is picked using the highest trajectory score. The training loss in Causal InfoGAN led to a latent model that most accurately represents possible changes to the rope, compared to the other two baselines.

## B.2 Binary States

In this work, we chose $P_\mathcal{M}(s)$ and $P_\mathcal{M}(a)$ to be fixed distributions, where each binary element was independent, with a Bernoulli$(0.5)$ distribution. In this case, the marginalization can be calculated in closed form. It is also possible to extend this model to a parametric distribution for $P_\mathcal{M}(s)$ and $P_\mathcal{M}(a)$, and marginalize using sampling.

## C Algorithm

Given the training data $\mathcal{D} = \{(o, o')|\ o \text{ and } o' \text{ are sequential observations.}\}$, Causal infoGAN learns a generative model that structures the latent space in a way that is useful for planning. We provide the algorithm details below:

Let $\theta_D, \theta_G, \theta_Q, \theta_T$ denote the parameters of neural networks $D, G, Q, T_\mathcal{M}$, respectively.

For a minibatch of $m$ samples $\{(o_i, o'_i)\}_{i=1}^m$ from $\mathcal{D}$, we:

- Generate $m$ fake samples

- Sample abstract states $s_1, \ldots, s_m$, where $s_i \sim P_{\mathcal{M}}$
- Sample next states $s'_1, \ldots, s'_m$, where $s'_i \sim T_{\mathcal{M}}(s'_i | s^i)$
- Sample noise $z_1, \ldots, z_m$, where $z_i \sim P_{\text{noise}}$
- Generate fake observations $\hat{o}_1, \hat{o}'_1, \ldots, \hat{o}_m, \hat{o}'_m$, where $\hat{o}_i, \hat{o}'_i = G(z_i, s_i, s'_i)$.

- Update the discriminator by descending its stochastic gradient

$$\nabla_{\theta_D} \left( -\frac{1}{m} \sum_{i=1}^{m} [\log D(o_i, o'_i) + \log(1 - D(\hat{o}_i, \hat{o}'_i))] \right)$$

- Update the generator and transition model by descending its stochastic gradient

$$\nabla_{\theta_G, \theta_T} \left( -\frac{1}{m} \sum_{i=1}^{m} [\log D(\hat{o}_i, \hat{o}'_i)] \right),$$

where the gradient $\theta_T$ is backpropagated using the reparametrization trick of Gumbel-softmax [16].

- Update posterior, generator, and transition model in the direction of maximal mutual information

$$\nabla_{\theta_Q, \theta_G, \theta_T} \left( \frac{1}{m} \sum_{i=1}^{m} [\log P_{\mathcal{M}}(s_i) - \log Q(s_i | \hat{o}_i) + \log T_{\mathcal{M}}(s'_i | s_i) - \log Q(s'_i | \hat{o}'_i)] \right),$$

where the gradient $\theta_T$ is backpropagated using the reparametrization trick of Gumbel-softmax [16].

- (*For continuous states with linear interpolation planning*) Update transition model to ensure small local transitions in the state space generate plausible observations.

$$\nabla_{\theta_T} \left( \frac{1}{m} \sum_{i=1}^{m} \|\Sigma_{\theta_T}(s_i)\|_2 \right),$$

where $\Sigma_{\theta_T}$ is part of $T_{\mathcal{M}}$ (see Section 4.3).

- (*Optional*) Update transition model to minimize a self-consistency loss (see details below)

$$\nabla_{\theta_T} \left( \frac{1}{m} \sum_{i=1}^{m} [-\log T_{\mathcal{M}}(s^*(o'_i) | s^*(o_i))] \right),$$

where $s^*(o) = \arg\max_s Q(s|o)$.

The self-consistency loss is added to further strengthen the relationship between transitions in the latent planning system and the real observations. We maximize the likelihood of observed transitions in the predicted states from real transitions. Namely, let $s^*(o) = \arg\max_s Q(s|o)$ denote the most likely state encoding for an observation, then the self-consistency loss is given by,

$$L_{sc}(\mathcal{M}) = \mathbb{E}_{o, o' \sim P_{\text{data}}} \left[ -\log T_{\mathcal{M}}(s^*(o') | s^*(o)) \right].$$

This loss guides $T$ to be consistent with $Q$ which stabilizes the training. We found this loss to help in stabilizing training for low-dimensional observations. We did not find that adding this loss is beneficial in the high-dimensional case, since in that case, while $Q$ provides meaningful state estimation on fake observations, it tends to overfit to the generated samples, and does not predict reliable states on real observations.

# D Experiment details

## D.1 2D Navigation Experiment

The model parameters we used for the toy domains is as follows:

In the key domain, we used a 4-dimensional space for the latent state[11]. Actions are sampled from a 3 dimensional space whereas the noise $z$ is 4 dimensional. The loss for the generator, the posterior and

the transition consistency are weighted equally with a learning rate of $10^{-4}$, whereas the learning rate for the discriminator is five times larger ($5 \times 10^{-4}$). We found that the transition consistency loss important in the stability of models using binary representations. The same hyperparameters are used for both the key domain and the $\epsilon$-key domain. In the latter $\epsilon = 0.1$.

In the tunnel domain we also used a 4-dimensional latent state (a 3-dimensional latent state gave similar results). Actions are sampled from a 3-dimensional space and noise is 4 dimensional. The learning rates are identical to those of the key domain.

To generate the training samples in the tunnel domain, the random walk had a characteristic length scale of 0.05. The rooms are from -1 to 1 in both width and height. The meridian is placed slightly off the middle, at $y = -0.1$. We bias the starting point in the particle trajectories around the choke in the middle, so that the sample trajectories have substantial probability is crossing from one room to the other.

In the key domain, we used a a characteristic length scale of 0.3. This much larger step size is needed because the particle needs to cover the top room, make it to the key zone (to obtain the key), and carry the key to the door to cross to the bottom room in a single trajectory.

In the tunnel domain, we chose the horizon $k$ to be uniform in $5 - 9$. That is, we sample observations $o, o'$ that are $k$-timesteps apart in the data when training the model. In the key domain, since the characteristic length scale is larger, we chose $k$ to be smaller, uniform in $1 - 4$.

In the key domain, we represent the possession of the key by a single number in the binary set $\{0, 1\}$. Incidentally, it was necessary to inject Gaussian noise to this key dimension during training. Otherwise the generator is required to learn a singularity around 0 and 1, making it numerically highly unlikely. We varied the normalized standard deviation of this Gaussian noise (w.r.t. $\epsilon$). Larger noise ($\leq 0.2$) produces more stable training, but too much noise can cause blurriness in the cluster boundaries. Overall the scale of this Gaussian noise doesn't substantially impact the representation that is learned.

The models are identical between the key domain and the tunnel domain. Both the generator, the discriminator and the binary posterior are two layer perceptrons with two 100 dimensional hidden layers. The transition function also has two hidden layers, with 10 neurons each.

For the represetantion of the latent space, we use a binary representation of the states as described in section 4.3. The generator uses a sequential architecture as described in Section 4.4, but the two outputs of the generator are trained with only 1 timestep in between with no autoregression on the autoregressive sub module.

## D.2 Rope Experiment

We use Adam [20] optimizer with the learning rate of 0.0002 for both discriminator and generator losses. The generator loss is the sum of three losses described in the Appendix C. We use coefficients 1 for the main generator loss, and 0.1 for both the mutual information and the transition loss. We deploy standard DCGAN architectures [34] for the discriminator D and the generator G. The posterior estimator Q has the same architecture as D with the change of the last CNN layer to output 128 channels and the addition of another layer of batchnorm, leaky ReLU and conv layer to the dimension of code. The details are described in table 3.

In DCGAN and infoGAN baselines, the size of latent code (or abstract state) and the noise is 7 and 2 respectively. In Causal InfoGAN, the generator takes in two abstract states at the same time so the size of noise is doubled to 4. However, we found that the result is quite robust to the dimension sizes.

The latent planning system uses a uniform prior between $[-1, 1]$ and a Gaussian transition with zero mean and state-dependent variance. The variance is diagonal and parametrized by a two-layer feed forward neural network of size 64 with ReLU nonlinearity. The last layer is exponentiated to output a positive value for the variance.

For training set, we use sequential observation pairs with 1 step apart from the rope dataset by Nair et al. [30].

### D.2.1 Causal Classifier

We trained a binary classifier to function as an evaluator for whether an observation transition is feasible or not, given the data. We use the classifier for two tasks: (1) To post-select transitions (in the observation space) during planning, and (2) to evaluate the score of a walkthrough trajectory.

During training, the classifier takes in a pair of images and output a binary classification of whether this image pair appears sequentially related. The training dataset consists of positive image pairs that are 1 timestep apart, and negative pairs that are randomly sampled from different rope manipulation runs. To avoid overfitting to the background in the rope dataset and learning a trivial solution where the classifier uses the background to distinguishi different runs, we preprocess the rope data using the background subtraction pipeline mentioned above.

The training accuracy converges to 100% on the training set, and 98% on a held-out test set.

To validate that this classifier actually learns to tell if the transition between two images is feasible or not, we evaluate it on images that are $k$ steps apart where the largest k is the length of an rope experiment. Despite the classifier never seeing samples that are more than 1 step apart, it learns to predict 0 probability for image pairs with large $k$. The prediction is well-behaved – As we increase k from 1 to the length of the run, the binary output smoothly and monotonically decreases from 1 to 0.

The model architecture used is a convolutional neural network with the following architecture. The two input images are concatenated channel-wise, fed together into the classifier. The optimization is done with the Adam optimizer with a learning rate of $10^{-3}$. These hyperparameters are not tuned since the performance of the classifier is sufficient.

| discriminator D / posterior estimator Q | generator G |
|---|---|
| Input 64 x 64 grayscale images (2 for D and 1 for Q) | Input a vector in $\mathbf{R}^7 \times \mathbf{R}^7 \times \mathbf{R}^4$ |
| 4 x 4 conv. 64 lReLU, stride 2 , batchnorm | 4 x4 upconv. 512 lReLU, stride 2 , batchnorm |
| 4 x 4 conv. 128 lReLU, stride 2 , batchnorm | 4 x4 upconv. 256 lReLU, stride 2, batchnorm |
| 4 x 4 conv. 256 lReLU, stride 2 , batchnorm | 4 x4 upconv. 128 lReLU, stride 2, batchnorm |
| 4 x 4 conv. 512 lReLU, stride 2 , batchnorm | 4 x4 upconv. 64 lReLU, stride 2, batchnorm |
| 4 x 4 conv. 1 for D and 128-batchnorm-lReLU-7 for Q | 4 x4 upconv. 2 Tanh (1 channel for each image) |

Table 3: The architectures for generating rope images. The discriminator takes in two grayscale images, and outputs the probability of the pair being real. The posterior shares the same architecture with D except the first and the last layer. It takes in one image, and outputs the mean and the variance of its predicted state. The generator takes in the current and the next abstract states (dim 7), and the noise (dim 4). It outputs the current and the next observations. The leaky coefficient is 0.2.

## Footnotes

[9]This is a general property of the entropy of a random variable, which only depends on the probability distribution and not on the variable values. In our case, for example, one can apply a permutation to the transition operator $T_{\mathcal{M}}$, and an inverse of that permutation to the generator's $s'$ input. Such a permutation would change the meaning of $s'$, without changing the mutual information term nor the distribution of generated observations.

[10]Note that in a system where the state is fully observable, the posterior is disentangled by definition, therefore in such cases the bound is tight.

[11]We also experimented with 3-5 dimensional latent spaces which gave similar results. Smaller latent space tend to have less expressive power and subject to generator collapse. Beyond 5 however the benefit is marginal.