[Reviews · NeurIPS 2018]

Reviewer 1



The paper proposes an approach to learn structured low-dimensional representations of a latent deterministic dynamical system using the realization of the system dynamics in the form of sample trajectories. The paper specifically tackles the 'reachable path' problem where the objective is to find a finite sequence of actions required to realize a given source-destination path with a priori fixed intermediate states. The authors introduce a new approach called the 'Causal InfoGAN' which I find to be a very minimalistic extension to the existing InfoGAN approach of Chen et.al. The authors have incorporated the self-consistency loss to the existing InfoGAN to quantify the correlation between the abstract states and the real observations. Further, I have the following concerns regarding the exposition of the paper. Problem 1 is irrelevant to the paper and so does not require explicit mentioning. Section 3.1 seems amateurish. Apart from the binary state representation (which is adapted from VaST) definition, I think the Bernoulli(0.5) details are inappropriate. Also the statement "is done using Gumbel-softmax trick for each element" should be elaborated. I have similar concerns for Section 3.2. It seems the section contains practical details and should be incorporated in the experimental section. The main content should be assigned to elucidating the conceptual aspects. The section does not add any value to the paper. In Section 4, the authors define their algorithm. The algorithm is very naively described. The algorithm seems to be the main contribution of the paper and it needs to described more vividly. Details like how the Causal InfoGAN optimization problem (4) is being realized tangibly is indeed lacking. In line 106-107, the authors mention that "balancing the complexity of the model and its accuracy". But I could not find details in the paper regarding how it is achieved. In lines 211-214, the details pertaining to training the Causal InfoGAN is lacking. The whole training procedure is explained in a single sentence. I believe more details are required to appreciate the algorithm. In line 238, the authors mention that the best trajectory is obtained as argmax min_{t} D(o_t, o_{t+1}). The details pertaining to the tractable approach to solve this particular step is not present. Overall, I find the paper very premature. Even though the empirical analysis shows considerable improvement (albeit debatable) which is indeed inspiring, I recommend the authors to reconsider the exposition of the paper to reflect the conceptual aspects with the experimental section corroborating it. ==== I have read the response, and would like to upgrade my score to 5 or 6. The current comparison study in the review is really not satisfactory.

Reviewer 2



1. Summary - This paper proposes Causal InfoGAN which is capable of generating a realistic sequence of observations from a given initial observation to a goal observation. The motivation is that this is solving a relaxed version of planning problem, which requires generating a sequence of actions. The idea of Causal InfoGAN is to train a generator that generates two consecutive observations given two consecutive “abstract states”, which encourages the model to generate realistic transitions. In order to enforce the consistency between abstract states and observations, this paper adds a mutual information maximization objective (which is the idea from InfoGAN). The experimental result on a toy 2D domain shows that the proposed model can learn reasonable abstract states that are consistent with the dynamics of the environment, as opposed to simple baselines (e.g., K-means) which are independent of the dynamics of the environment. The qualitative result on Rope Manipulation data shows that Causal InfoGAN can generate more realistic transitions than InfoGAN. [Pros] - Novel application of GAN that is capable of generating a sequence of observations that is consistent with the dynamics of the environment. - The proposed idea is sensible and interesting. [Cons] - The experimental result is okay but not much comprehensive. 2. Quality - The idea of training a GAN to generate realistic transitions that is interesting. - The experimental result could be more comprehensive. First, the paper only presents qualitative results for Rope Manipulation task. Though Casual InfoGAN looks better than InfoGAN, some transitions seem very unrealistic. It would be better to provide quantitative results (something like Table 1). Second, it would be much convincing/interesting to show results on more complex and popular domains such as Atari and MuJoCo. Lastly, it would be good to show an evidence that the learned model can generalize to unseen observations, which would be one of the advantages of model-based RL. - It would be good to discuss the following work in the related work section: Universal planning network [Srinivas et al.] and Model-Based Planning with Discrete and Continuous Actions [Henaff et al.]. They seek to generate a sequence of “actions” given initial/goal observations. 3. Clarity - The problem formulation and the description of the method are well-written. - In Section 2, the paper claims that the planning problem (generating a sequence of actions) is “unnecessarily” difficult to solve, in order to justify their relaxed problem (generating observations). This sounds like a too strong argument. How can we build a control agent if we do not care about generating actions and only generate “observations”? 4. Originality - The idea of learning an abstract transition model using GAN is novel to my knowledge. 5. Significance - I would personally view this work as an interesting application of GAN for dynamical systems rather than “learning plannable representation” that sounds quite ambitious, because the authors are not tackling the full planning problem (e.g., dealing with actions, exploration, reward). Nevertheless, I think this paper is tackling an interesting problem with a new idea, which can be potentially useful for control in the future.

Reviewer 3



This paper attempts to learn planable representations of dynamical systems. The authors propose a clever modification of the InfoGAN to capture causal connections; this results in an "abstract state space" where causally relevant information is captured by the abstract states, and which is then used to break down complex planning problems into smaller, manageable chunks. I really liked this paper, although a major weakness is the experimental evaluations. On the positive side: I *love* the idea of learning a low-dimensional manifold of abstract states that capture causally relevant information. I felt that the causal extensions to the InfoGAN, while not an earth-shattering advance, were a solid idea that seems elegant and likely to work. More generally, I feel that learning planable abstract representation is the Right Thing To Do, so I'm glad to see a well-executed version of the idea. The paper is well-written, easy to read, has all the right amounts of detail, etc. On the negative side: The experiments were a big disappointment. I was so excited as I read the paper, anxiously awaiting the rope manipulation experiment. But to be honest, the result wasn't that impressive -- while I suppose it's true that Fig. 3(a) is better than Fig. 3(b), I don't consider this to be really close a "solution". The fact that there was no real comparison on the rope task was also a concern. The multi-room experiment seemed a bit contrived, but it made the point. Overall, I'm was bit torn about this paper, but I ultimately decided that the strength of the framework and the ideas is worth considering as a NIPS publication. I base that assessment largely on the fact that as I was reading the paper, I had several ideas come to me where I felt like I could build on the results -- hierarchical versions, object-oriented versions, temporally extended versions. This is typically the mark of a solid idea. So, we'll see where it lands in the pile of NIPS submissions. But at any rate, I strongly encourage the authors to pursue this most interesting line of work!

Reviewer 4



The article presents Causal InfoGAN, a framework for learning a low-dimensional generative model and structured representations of sequential high-dimensional observations, such as images, from a dynamical system, for use in long-horizon planning. The focus is discrete and deterministic dynamics models that can be used in graph search methods in, for example, classical AI planning The work follows the InfoGAN approach, adding to the GAN training a loss function that maximizes the mutual information between observation pairs and transitions in the planning model. The GAN is trained to generate sequential observation pairs from the dynamical system maximizing the mutual information between the generated observations. The objective helps the abstract model to capture the relevant changes in the observations, for simplicity this work ignores partial observability. The generative process is induced by a transition in a low-dimensional planning model with noise such that "small changes to the abstract state correspond to feasible changes to the observation" -- planning in this model involves "interpolating between abstract states". The transitions in the planning model lead to a low-dimensional representation that is claimed to explain the causal nature of the data. The experiments reported using Causal InfoGAN include * A toy 2D navigation problem which shows that state abstraction leads to better planning, with clusters that correspond to the possible dynamics of the particle in the task that leads to reasonable generated planning trajectories. The model does not rely on the Euclidean metric which leads to better planning -- the authors point out that E.D can be unsuitable even on low-dimensional continuous domains. * A Rope manipulation data problem which shows that training loss of Causal InfoGAN can generate a latent space that leads to accurate representation of possible changes to the rope and generates high-level plans directly from data without requiring additional human guidance. It is claimed that the objective guarantees feasible transitions and "causality preserving loss" leads to a learned latent space in which local perturbations correspond to plausible changes in the image (unlike an infoGAN with a MI loss that does not involve state transitions). Causal InfoGAN appears to generate convincing "walkthrough sequences" for manipulating a rope into a given shape, using real image data collected from a robot randomly manipulating the rope. The paper is generally clear and well written, the quality of the work and the paper seems to be reasonably good, to my knowledge the work seems original and its relation to other work in the field seems to be clearly described, I do not feel able to comment on the significance of the work.